# Deep-Learning-Based Smartphone Application for Self-Diagnosis of Scleral Jaundice in Patients with Hepatobiliary and Pancreatic Diseases

**DOI:** 10.3390/jpm11090928

**Published:** 2021-09-18

**Authors:** Joon Hyeon Park, Min Jae Yang, Ji Su Kim, Bumhee Park, Jin Hong Kim, Myung Hoon Sunwoo

**Affiliations:** 1Department of Electrical and Computer Engineering, Ajou University, Suwon 16499, Korea; junpark25@ajou.ac.kr; 2Department of Gastroenterology, Ajou University School of Medicine, Suwon 16499, Korea; creator1999@hanmail.net; 3Office of Biostatistics, Medical Research Collaborating Center, Ajou Research Institute for Innovation, Ajou University Medical Center, Suwon 16499, Korea; k.jisu5107@aumc.ac.kr (J.S.K.); bhpark@ajou.ac.kr (B.P.); 4Department of Biomedical Informatics, Ajou University School of Medicine, Suwon 16499, Korea

**Keywords:** deep-learning, hepatobiliary disease, outpatient care, pancreatic disease, scleral jaundice, self-diagnosis, serum hyperbilirubinemia, smartphone application, total bilirubin levels

## Abstract

Outpatient detection of total bilirubin levels should be performed regularly to monitor the recurrence of jaundice in hepatobiliary and pancreatic disease patients. However, frequent hospital visits for blood testing are burdensome for patients with poor medical conditions. This study validates a novel deep-learning-based smartphone application for the self-diagnosis of scleral jaundice in such patients. The system predicts total serum bilirubin levels using the deep-learning-based regression analysis of scleral photos taken by the smartphone’s built-in camera. Enrolled patients were randomly assigned to either the training cohort (*n* = 90, 1034 photos) or the validation cohort (*n* = 40, 426 photos). The intraclass correlation coefficient value for predicted serum total bilirubin (PSB) derived from the images repeatedly taken at the same time for the same patient showed good reliability (0.86). A strong correlation between measured serum total bilirubin (MSB) and PSB was observed in the subgroup with MSB levels ≥1.5 mg/dL (Spearman rho = 0.70, *p* < 0.001). The receiver operating characteristic curve for PSB showed that the area under the curve was 0.93, demonstrating good test performance as a predictor of hyperbilirubinemia (*p* < 0.001). Using a cut-off PSB ≥1.5, the prediction sensitivity of hyperbilirubinemia was 80.0%, with a specificity of 92.6%. Hence, the tool is effective for patient monitoring.

## 1. Introduction

Jaundice is a yellow discoloration of body tissues caused by the deposition of bilirubin, a product of the breakdown of heme. It occurs in the presence of serum hyperbilirubinemia when there is an imbalance between bilirubin production and clearance in hepatobiliary and pancreatic diseases or, less often, hemolytic disease. Jaundice begins to appear on the skin from 1.5 mg/dL of serum total bilirubin in the blood, and it is best detected by examining the sclera, which has a particular affinity for bilirubin because of its high elastin content [1,2]. However, scleral jaundice can, in fact, be recognized with the naked human eye when the total serum bilirubin level is above 2–3 mg/dL [3,4]. Hence, periodic outpatient-based surveillance with blood testing has been indispensable to the early detection of hyperbilirubinemia in patients with hepatobiliary and pancreatic diseases. Unfortunately, traveling to clinics can be burdensome for patients with poor medical conditions related to advanced cancers or disabling comorbidities.

As the mobile healthcare industry is rapidly expanding, biometric information collected by mobile devices is being used for self-disease screening and telemedicine in various medical fields. Currently, information processed by mobile devices is further objectified by artificial intelligence technologies. Advanced scientific progress has thus influenced the patterns of healthcare utilization [5,6].

Recently, an image-based mobile application was developed for early, non-invasive screening of scleral jaundice as an alternative to blood tests. It predicts serum total bilirubin levels using deep-learning-based regression analysis of scleral photos taken by the patient’s smartphone camera. In the current study, we validated this deep learning-based smartphone application for the self-diagnosis of scleral jaundice in patients with hepatobiliary and pancreatic diseases.

## 2. Materials and Methods

### 2.1. Patients

From August 2018 to August 2019, patients with current or past benign or malignant hepatobiliary and pancreatic diseases over 20 years of age were prospectively enrolled at Ajou University Hospital, Korea. To ensure the reliability of this mobile device, both currently icteric and anicteric patients were included in this study. We excluded patients having abnormal scleral features, such as dots or wounds. This study was approved by the Institutional Review Board of Ajou University Hospital (approval number: AJIRB-DEV-SUR-18), and informed consent was obtained from each patient.

### 2.2. Data Collection

The dataset of each patient comprised digital photo images of the patient’s eye and measured serum total bilirubin (MSB) levels from the patient’s blood sample. Images were taken within 24 h before or after a blood test.

The process of photographing the patients’ sclera using a mobile device is shown in Figure 1a. An accessory color consistency patch was used for color adjustment because different lighting conditions can change the colors of the same scene. The patch was made of a sheet of white paper (9 × 5 cm) with a rectangular hole on the upper part to catch the patient’s eye area and a black and white square line structure on the lower part for color adjustment. After centering the patient’s eye in the hole in the upper part of the patch, four photos were consecutively taken with the smartphone’s built-in camera (Samsung Galaxy S8 smartphones, Samsung Electronics Co., Ltd., Suwon, Korea) without flash and saved in the raw format. Thereafter, the application analyzed the scleral images using the built-in deep-learning system and presented the predicted serum total bilirubin (PSB) as an output.

### 2.3. Architecture of Deep-Learning System (Figure 1b)

The deep-learning system comprises a convolutional neural network (CNN) that uses a function that includes nearby pixels to perform feature extraction and color correction of the scleral image [7]. A CNN is a type of artificial neural network that uses functions that compute information about nearby pixels for deep learning and has been applied to visual image analysis in many areas [8,9,10]. The deep-learning system derives the PSB value using the pixel information of the scleral smartphone image. The training stage applied input scleral images and actual MSB information. During the training process, the neural network parameters were set randomly, and the parameters were modified to gradually reduce the error by comparing the PSB value for each sclera image derived from the deep-learning system and the actual MSB value. These processes were repeated for all images included in the training set, and the neural network learned to predict the MSB from patient scleral smartphone images. The effectiveness of the deep-learning system was assessed by using hold-out cross-validation, in which the dataset was randomly divided into training and validation sets in a 7:3 ratio.

### 2.4. Outcome Measurements and Statistical Analyses

MSB is the total bilirubin value measured from the patients’ peripheral blood within 24 h before and after sclera imaging, whereas the PSB is the total bilirubin value predicted by the deep-learning system built into the application when the sclera photo information is transmitted to the smartphone application. In this study, hyperbilirubinemia requiring clinical attention or intervention is defined as an elevated MSB level above 1.5 mg/dL.

The current study first analyzed the reproducibility among the PSBs derived from consecutive scleral smartphone photos of the same patient repetitively taken at a specific time, evaluated using the intraclass correlation coefficient (ICC), which is categorized as follows: an ICC value of <0.5 is poor, 0.5–0.75 is moderate, 0.75–0.9 is good, and >0.9 is excellent [11]. Then, the correlation between the MSB and TSB was analyzed using Spearman’s rank-order correlation coefficient (Spearman’s rho), which is categorized as follows: <0.20 is negligible, 0.20–0.39 is weak, 0.40–0.69 is moderate, 0.7–0.89 is strong, and ≥ 0.9 is very strong [12]. Finally, the receiver operating characteristic (ROC) curve was computed to test the strength of the PSB derived from the deep-learning-based mobile device and to determine the predictive cut-off value of PSB for detection of hyperbilirubinemia. The PSB value with balanced high sensitivity and specificity for the diagnosis of actual hyperbilirubinemia was selected as the cut-off value for prompting a visit to an outpatient clinic for blood testing.

All statistical analyses were performed using R software v.3.6.3 (R Development Core Team, Vienna, Austria). Statistical significance was set at *p* < 0.05.

## 3. Results

During the study period, 130 patients were included in this study, and a total of 1460 sclera photos were taken in all study populations, with an average of 2.3 hospital visits per patient. Enrolled patients were randomly assigned to either the training cohort (*n* = 90, 1034 photos) or the validation cohort (*n* = 40, 426 photos) in a 7:3 ratio. The baseline characteristics of the patients are summarized in Table 1. There were no significant differences in patient demographics and laboratory data between the training and validation cohorts.

The ICC value for PSBs derived from the deep-learning analysis of the scleral smartphone images repetitively taken at the same time for the same patient showed good reliability with a value of 0.86 (95% confidence interval (CI) 0.82–0.90, *p* < 0.001). There was a moderate correlation between MSB and PSB (Spearman rho = 0.59, *p* < 0.001), whereas a strong correlation between MSB and PSB was observed in the subgroup with MSB level ≥1.5 mg/dL (Spearman rho = 0.70, *p* < 0.001) (Figure 2). The ROC curve for PSB showed that the area under the curve was 0.93 (95% CI, 0.90–0.96), demonstrating a good test performance as a predictor of hyperbilirubinemia (*p* < 0.001) (Figure 3). The optimal threshold value of PSB for predicting hyperbilirubinemia was 1.58, which showed a balanced high sensitivity and specificity. The cut-off value was set at 1.5 for clinical convenience. Using a cut-off PSB value equal to or greater than 1.5, the sensitivity to predict hyperbilirubinemia was 80.0% (95% CI, 0.72–0.87) with a specificity of 92.6% (95% CI, 0.89–0.95) (Table 2). The overall accuracy was 89.2% (95% CI, 0.86–0.92).

## 4. Discussion and Conclusions

Jaundice is often the first manifestation of hepatobiliary and pancreatic diseases. It gradually improves with a decrease in total serum bilirubin following definitive treatment or palliative biliary drainage, whereas recurrence or exacerbation of the underlying diseases elevates serum bilirubin again, leading to recurrent jaundice. Therefore, early detection of recurrent hyperbilirubinemia provides patients with the opportunity to receive prompt therapeutic intervention, thus minimizing disease progression-related morbidity or mortality.

In routine outpatient follow-ups for patients with current or prior hepatobiliary and pancreatic diseases, hyperbilirubinemia is screened using blood tests because scleral jaundice is difficult to recognize with the naked eye when total serum bilirubin is less than 3 mg/dL. However, frequent hospital visits for blood sampling, even in the absence of specific symptoms, are burdensome for patients with financial difficulties related to long-standing struggles against the disease and those who suffer from impaired mobility related to advanced cancer or disabling comorbidity. Hence, there is an unmet need for non-invasive and easy-to-access diagnostic methods that can replace blood testing while reliably detecting scleral jaundice, even when the bilirubin level ranges between 1.5 and 3 mg/dL [3,4].

The mobile healthcare industry is rapidly growing because of the growing interest in health monitoring using smartphone development. Furthermore, biometric information can be measured using various sensors mounted on a smartphone (e.g., camera, microphone, gyroscope, accelerometer, fingerprint scanner) or wearable devices linked to the smartphone. Based on the results of collecting and analyzing measured biometric information, the mobile healthcare industry has spawned many types of applications, including telemedicine, self-diagnosis, and healthcare [7].

BiliCam [13] was the first smartphone-based medical device for monitoring newborn jaundice in the skin. The smartphone’s built-in camera and a color calibration card were used to mitigate the effects of different lighting conditions. Before taking photos, the calibration card was placed on the belly of the newborn. Based on the color information derived from images captured by a smartphone, BiliCam predicted MSB levels using a machine learning-based regression algorithm. In a study based on 100 newborns, the PSB levels by BiliCam correlated with the MSB by a linear correlation coefficient of 0.84, showing a moderate correlation. However, in the clinical field, yellow skin discoloration on the belly is rarely detected until the serum bilirubin level is at least 7–8 mg/dL. Therefore, there is still a demand for new applications that can diagnose jaundice in the sclera or mucous membrane, where jaundice is detectable even at a lower level of serum bilirubin.

Biliscreen [14] was the first smartphone-based system to predict a patient’s bilirubin level by analyzing scleral photos. Unlike BiliCam, it quantified the extent of jaundice in the adult sclera and used a goggle box to control the amount of light that reaches it. After taking scleral photos, scleral segmentation from the images was performed manually. Using machine-learning-based random-forest regression, Biliscreen predicted the MSB and achieved a Pearson correlation coefficient of 0.89 in a 70-person clinical study. Despite their favorable performances, BiliCam and Biliscreen were mainly limited by their small study populations and imperfect color-calibration methods.

At the beginning of our study, we attempted to determine the correlation between the yellowness of the sclera and the level of MSB by analyzing the average hue, saturation, and value (HSV) of the sclera. However, HSV color scales of the same sclera were significantly influenced by different external lighting conditions; thus, the correlation between MSB and the average HSV scale of the sclera was inconsistent, even in the same patient. To compensate for this confounding factor, we used an accessory patch for color constancy to make things that are white in person appear white in photos. Although this attempt was to adjust the color by the patch, color calibration was still problematic because color stimuli from adjacent pupils, veins, tears, or reflected light were complex and varied according to the patient and could not be perfectly controlled. Therefore, we applied a deep-learning algorithm to overcome this obstacle.

Our smartphone application is the first system to diagnose jaundice by analyzing biometric information measured using mobile technology based on a deep-learning algorithm. With this application, all color information of the sclera and adjacent confounding structures are simultaneously analyzed using a deep-learning algorithm, which provides the PSB levels for each patient.

In this study, the PSB showed good reproducibility in the same patient and good test performance as a predictor of hyperbilirubinemia. Likewise, the correlation between MSB and PSB in patients with MSB levels ≥ 1.5 mg/dL was strong, although it was limited when MSB levels were >1.5 mg/dL. For clinical application, we set a cut-off value with balanced sensitivity and specificity. At the PSB cut-off value of 1.5, the sensitivity for diagnosing real hyperbilirubinemia was 80%, which is somewhat less than expected. The sustainable development of applications, including larger-scaled training and validation cohorts for deep-learning analysis, is needed to improve the correlation efficacy and sensitivity. Moreover, in the current study, we evaluated the feasibility of this application exclusively on the smartphone system from only one manufacturer. To generalize the measured outcomes of the study, a future study based on various smartphone models is needed.

Similar to BiliCam and Biliscreen, an accessory color constancy patch adjustment tool was used in this study. To improve patient convenience by simplifying the smartphone-based self-diagnosis process, it is necessary to develop a new technology in which the color adjustment process automatically operates via the smartphone’s built-in camera without an accessory.

In conclusion, smartphone-based diagnosis of scleral jaundice using a deep-learning system is useful as an alternative to serum bilirubin testing during patients’ follow-ups related to hepatobiliary and pancreatic diseases. To apply this deep-learning-based mobile device in the real clinical field, there is a need for further development of deep-learning algorithms with various mobile devices and feedback through multicenter-based large-scale training and validation cohorts.

## Figures and Tables

**Figure 1 jpm-11-00928-f001:**
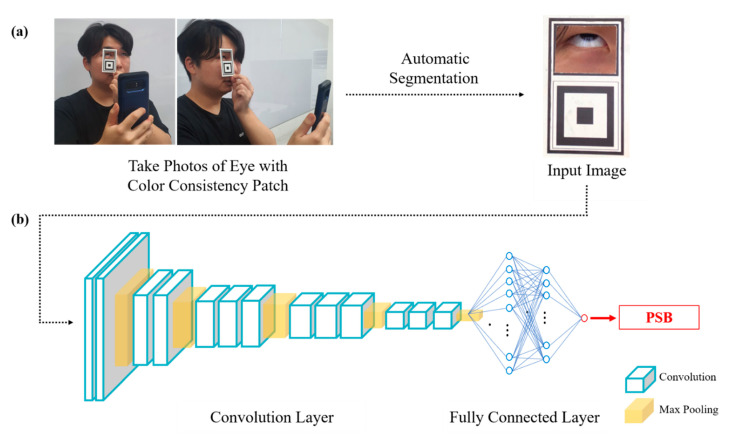
Process of deriving predicted serum total bilirubin (PSB) by analyzing scleral photos taken with a mobile device using the deep-learning system: (**a**) photographing the patients’ sclera using a mobile device and color consistency patch; (**b**) analysis of the scleral images using the convolutional neural network of the built-in deep learning system for total predicted serum bilirubin.

**Figure 2 jpm-11-00928-f002:**
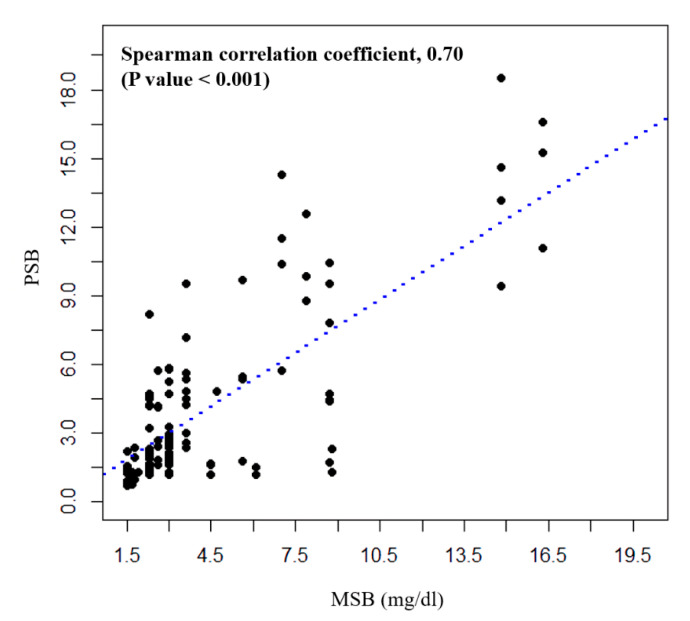
Correlation between total measured serum bilirubin (MSB) and total predicted serum bilirubin (PSB) in patients with hyperbilirubinemia (MSB level ≥ 1.5 mg/dL).

**Figure 3 jpm-11-00928-f003:**
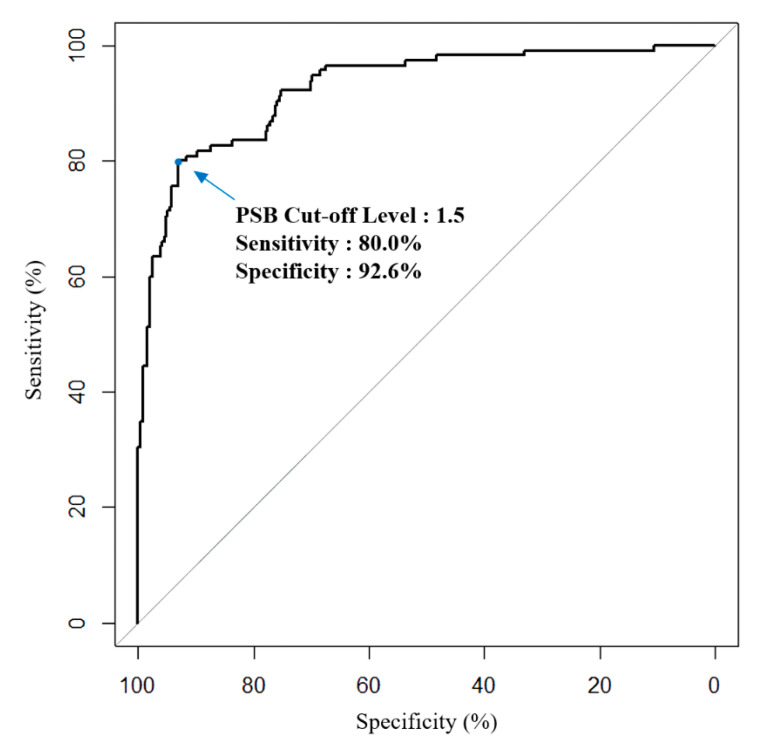
Receiver operating characteristic curve for the total predicted serum bilirubin (PSB) level derived from the deep-learning system of the smartphone application as a predictor of actual serum hyperbilirubinemia. The cut-off level of 1.5 is useful to predict serum hyperbilirubinemia with a sensitivity of 80.0% and specificity of 92.6%.

**Table 1 jpm-11-00928-t001:** Baseline characteristics of training and validation sets.

Characteristics	Total	Training Set	Validation Set	*p*-Value
No. of patients	130	90	40	
No. of hospital visits	298	211	87	
Average visit per patient	2.3	2.3	2.1	0.67
No. of total photos	1460	1034	426	
Photos/cases	4.9	4.9	4.9	0.95
Age	70.0 ± 4.8	68.9 ± 15.4	72.6 ± 13.1	0.05
Male (%)	173 (58.1)	120 (56.9)	53 (60.2)	0.51
Underlying disease				0.17
Hepatobiliary	92 (70.8)	67 (74.4)	25 (62.5)	
Pancreatic disease	38 (29.2)	23 (25.6)	15 (37.5)	
Total bilirubin	0.8 (0.4–1.8)	0.9 (0.5–1.9)	0.7 (0.4–1.6)	0.88
AST	31.0 (19.0–70.0)	31.0 (19.0–70.0)	31.5 (21.3–67.8)	0.49
ALT	19.0 (9.0–50.0)	20.0 (9.0–54.0)	18.5 (9.0–41.5)	0.21

Data are presented as mean ± SD, number (%), and median (interquartile range). AST: aspartate aminotransferase; ALT: alanine aminotransferase; NS: not significant.

**Table 2 jpm-11-00928-t002:** Cut-off predicted serum bilirubin levels for predicting actual serum hyperbilirubinemia.

PSB Level	Sensitivity	Specificity	PPV	NPV
1.3 mg/dL	82.6%	83.6%	65.1%	92.9%
1.5 mg/dL	80.0%	92.6%	80.0%	92.6%
2.0 mg/dL	63.4%	97.1%	89.0%	87.8%
2.5 mg/dL	52.2%	98.1%	90.9%	84.7%

PPV, positive predictive value; NPV, negative predictive value; AUC, area under the curve.

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
