# Peer review of "Deep-Learning-Based Smartphone Application for Self-Diagnosis of Scleral Jaundice in Patients with Hepatobiliary and Pancreatic Diseases"

_jpm, 2021, doi:10.3390/jpm11090928_

Round 1

Reviewer 1 Report

This study is very original and well planned. Through the clinical use of this new application, I hope that artificial intelligence-related equipment will be widely utilized in the field of hepatobiliary and pancreatic. It would be better to do some minor revisions.

1. In Table 1, total bilirubin, AST, and ALT levels are recommended to be resented as median and interquartile ranges.

2. Figure 2. Both the horizontal and vertical axes represent bilirubin levels, so it is better to express them in square (not rectangle).

Author Response

Comment #1: In Table 1, total bilirubin, AST, and ALT levels are recommended to be resented as median and interquartile ranges.

Response #1: We sincerely appreciate your thoughtful comments. The total bilirubin, AST, and ALT levels have been presented as median and interquartile ranges in Table 1.

Comment #2: Figure 2. Both the horizontal and vertical axes represent bilirubin levels, so it is better to express them in square (not rectangle).

Response #2: We have revised Figure 2 to a square.

Reviewer 2 Report

This is an interesting analysis of deep-learning-based smartphone application for self-diagnosis of scleral jaundice in patients with hepatobiliary and pancreatic diseases. The authors demonstrate clinical relevant sensitivity and specificity rates for patient monitoring using the described method.

The paper is clearly written and the experimental procedures and statistics are sound and valid. There are only few minor points.

One drawback is that only one smartphone/ camera system was used. Could the authors comment?

How was the size of the training and validation cohort determined?

Author Response

Responses to the comments of Reviewer 2

Comment #1: One drawback is that only one smartphone/ camera system was used. Could the authors comment? 

Response #1: We completely agree that using a single type of smartphone/camera system can be a major drawback of this study. In the near future, we will evaluate the feasibility of this application on various smartphone models through a well-designed, larger-scale study.

To reflect your valuable comment, we have mentioned this limitation in the discussion section of the revised manuscript as follows:

“Moreover, in the current study, we evaluated the feasibility of this application exclusively on the smartphone system from only one manufacturer. To generalize the measured outcomes of the study, a future study based on various smartphone models is needed.”

Comment #2: How was the size of the training and validation cohort determined?

Response #2:

In this study, the effectiveness of the deep learning system was assessed using a hold-out cross-validation in which the dataset was randomly divided into training and validation sets. In the hold-out method, the size of each set is arbitrary, but 8:2 and 7:3 ratios are widely used as the ratios of training set to validation set. Therefore, we assigned the enrolled patients to either the training and validation sets in a 7:3 ratio.

To address your comment, the information about the hold-out cross validation has been added to Section 2.3 (Architecture of Deep-learning System) in the revised manuscript as follows:

“The effectiveness of the deep-learning system was assessed by using a hold-out cross-validation in which the dataset was randomly divided into training and validation sets in a 7:3 ratio.”
